# High-Throughput Griess Assay of Nitrite and Nitrate in Plasma and Red Blood Cells for Human Physiology Studies under Extreme Conditions

**DOI:** 10.3390/molecules26154569

**Published:** 2021-07-28

**Authors:** Andrea Brizzolari, Michele Dei Cas, Danilo Cialoni, Alessandro Marroni, Camillo Morano, Michele Samaja, Rita Paroni, Federico Maria Rubino

**Affiliations:** 1Laboratory for Analytical Toxicology and Metabonomics, Department of Health Sciences, Università degli Studi di Milano, v. A. di Rudinì 8, 20142 Milan, Italy; andreabrizzolari79@gmail.com; 2DAN Europe Research Division, Contrada Padune, 64026 Roseto degli Abruzzi, Italy; dcialoni@daneurope.org (D.C.); amarroni@daneurope.org (A.M.); 3Laboratory of Clinical Chemistry and Mass Spectrometry, Department of Health Sciences, Università degli Studi di Milano, v. A. di Rudinì 8, 20142 Milan, Italy; michele.deicas@unimi.it (M.D.C.); camillo.morano@studenti.unimi.it (C.M.); rita.paroni@unimi.it (R.P.); 4Laboratory of Biochemistry, Department of Health Sciences, Università degli Studi di Milano, v. A. di Rudinì 8, 20142 Milan, Italy; michele.samaja@unimi.it

**Keywords:** Antarctica, diving, extreme environment, hypoxia, hyperbaria, maladaptation, nitric oxide, nitrosative stress, underwater activity

## Abstract

The metabolism of nitric oxide plays an increasingly interesting role in the physiological response of the human body to extreme environmental conditions, such as underwater, in an extremely cold climate, and at low oxygen concentrations. Field studies need the development of analytical methods to measure nitrite and nitrate in plasma and red blood cells with high requirements of accuracy, precision, and sensitivity. An optimized spectrophotometric Griess method for nitrite–nitrate affords sensitivity in the low millimolar range and precision within ±2 μM for both nitrite and nitrate, requiring 100 μL of scarcely available plasma sample or less than 50 μL of red blood cells. A scheduled time-efficient procedure affords measurement of as many as 80 blood samples, with combined nitrite and nitrate measurement in plasma and red blood cells. Performance and usefulness were tested in pilot studies that use blood fractions deriving from subjects who dwelt in an Antarctica scientific station and on breath-holding and scuba divers who performed training at sea and in a land-based deep pool facility. The method demonstrated adequate to measure low basal concentrations of nitrite and high production of nitrate as a consequence of water column pressure-triggered vasodilatation in deep-water divers.

## 1. Introduction

Nitric oxide (NO) was identified in the late 1980s as the first of the “gasotransmitter” class of small-molecule endogenous mediators that are biochemically generated, diffuse, and specifically bind sensing macromolecules for signaling purposes within the body. While NO acts primarily as an endogenous vasodilator, soon an increasing variety of other physiological functions emerged [1].

The human vascular system, when experiencing several extreme conditions, such as cold temperature, hypoxia, low or high atmospheric pressure, triggers survival responses to cope with the action of exogenous stressors, which include modulation of the vascular flow mediated by the action of NO [2,3].

NO plays a key role in subjects exposed to high hydrostatic pressures (i.e., the one of water during diving) [4], and recent measurements taken in SCUBA divers and breath-hold divers at −40 m depth showed remarkable increases in the plasma concentrations of NO derivatives [5,6]. Breathing air at hyperbaric conditions raises oxygen partial pressure (pO_2_), leading to hyperoxia and causing vasoconstriction and oxidative stress [7,8], which is at the base of endothelial dysfunction. Diving-related Venous Gas Emboli (DVGE) is frequently observed in divers and may be a way to explain endothelial dysfunction after underwater activities [9], with particular regard to NO-related endothelial changes.

To measure the pool of nitrogen oxides, several methods have been employed. NO is a radical with a short biological half-life. This property makes its direct measurement in body fluids very difficult, especially during field studies where an equipped laboratory is not readily available [10]. Thus, it is more reliable to measure, as a proxy of the biological pool of NO, its stable metabolic products [11], nitrite and nitrate, and express the results either as total nitrogen oxides (NOx) or as the separate anions.

Although several techniques can be used to measure nitrite, the most frequently employed is the century-old Griess reaction, which is a staple of most dye chemistry. The reaction of nitrite with an aromatic amine yields the corresponding diazonium ion, which, in turn, reacts with another aromatic amine or phenol to yield a diazo compound (reaction scheme of Figure 1) [12].

With an appropriate choice of the aromatic compounds, intensely colored “azo-dyes” are obtained. The color development is fast, and the reaction can be elaborated into a very sensitive, simple, and cost-effective analytical method to detect and measure nitrite, mostly in surface, drinking, and wastewater [13,14]. Traditional acceptor amines, such as alpha-naphtylamine, are carcinogenic and poorly water-soluble and have been superseded by less hazardous ones, such as the ethylene-diamine derivative depicted in the reaction scheme.

Nitrate can be measured with the same method after its stoichiometric reduction to nitrite [15,16,17]. This reaction can be accomplished with several inorganic reagents and with a specific natural enzyme that uses NAD(P)H as the reducing co-factor [18]. In this case, both nitrite (if any present) and nitrate are measured in the sample, and speciation can be achieved by separately measuring nitrite without and following the reduction of sample nitrate [15,16,17,19]. Due to the necessity to operate the subtraction between two error-prone analytical measurements, analytical uncertainty needs to be controlled in both analytical steps of nitrate reduction and color development.

The Griess method has been adapted to the measurement of nitrite/nitrate in biological samples, such as plasma/plasma, tissue homogenates, and urine [11,18]. Pre-analytical sample preparation, especially the elimination of proteins, which interfere with the assay, is necessary for obtaining robust results.

In biomedical research, while the measurement of nitrite and nitrate in plasma is most common, red blood cells are becoming increasingly attractive for physiological studies that involve circulation and the response to environmental stressors. The red blood cell is extremely sensitive to oxidative stress; thus, the measurement of the nitrite/nitrate ratio in this condition entails a complementary informative value.

Here, we present an optimized and validated procedure to measure nitrite and nitrate in the protein-free extract of plasma and erythrocytes and a fitness-for-purpose test of its application to our current studies. The analytical results presented in the fitness-for-purpose section were derived from blood samples taken from subjects who participated in extreme environment studies [5,6]. One is dwelling for six months at an Antarctica remote exploration facility (the Concordia Research Station, at http://www.concordiastation.aq/home-1/; last accessed on 27 July 2021) [20]. The other is from divers who were sampled in different conditions, including in a training facility and/or in the open sea [21]. Both are examples of contemporary physiological research aimed at studying the response of the organism to physiological extremes of temperature cycling, external pressure, and exposure to varying concentrations of inhaled oxygen concentration.

## 2. Results

### 2.1. Optimization of Reaction Parameters

#### 2.1.1. Color Development and Nitrite Concentration Measurement

The measurement of nitrite and total nitrate in the samples with the Griess method entails the incorporation of one nitrogen atom of the nitrite pool in one molecule of an azo dye, according to the reaction scheme in Figure 1.

The reference reaction was performed on water nitrite solutions at concentrations in the 5–200 μM range expected for the levels in plasma and in red blood cells. Color development was near instantaneous, and complete stabilization of the absorbance occurred within 20 min after the addition of 75 mM sulphanylamide and 3.86 mM *N-*(1-naphthyl)ethylenediamine. A higher temperature (55 °C), or longer reaction time of 50 min resulted in a degradation of method performance, with a higher Limit-of-Detection (LoD) (2.3 ± 0.8 μM) and a higher standard error (SE) (1.6 ± 0.6 μM), with respect to the reaction at ambient temperature (20–25 °C) and 20- or 15-min reaction time (LoD 1.0 ± 0.5 μM; SE 0.7 ± 0.4 μM). A reaction time of 15 min at ambient temperature was deemed sufficient to achieve the most stable results.

#### 2.1.2. Nitrate to Nitrite Reduction and Total Nitrogen Oxides Measurement

The chemical reduction of nitrate to nitrite with vanadyl ion in solution is well established for the preparation of water samples to measure nitrate with the Griess method. Reagent concentration, reaction temperature, and time are the most relevant parameters to control, especially when the sample contains both nitrate and nitrite. Over-reaction (high temperature, extended reaction time, or an unnecessarily high concentration of vanadyl) leads to a reduction of nitrite to NO and its loss by evaporation from the sample. Reduction conditions strongly depend on the sample matrix and should therefore be optimized along with the pre-analytical sample treatment.

Control solutions of vanadyl showed a high concentration-dependent absorbance in the blank. A kinetic study targeted conditions to obtain the closest-possible slope value to that of nitrite with the lowest-possible value of intercept absorbance by modulating the reductant concentration, reaction temperature, and time.

A reduction at room temperature did not proceed at any meaningful velocity; therefore, temperatures up to 60 °C were investigated due to mechanical strength limitations of the polystyrene sample plate at higher temperature and to irregular evaporation of the liquid from the wells, even using the covering lid. A temperature of 55 °C was deemed an optimal compromise for practical work.

The reductant concentration and reduction time were optimized in a further study. Two reductant concentrations were investigated: 25 mM and 12.5 mM. The higher excess of reductant entailed a more variable value of intercept absorbance of the blank (zero-concentration in the standard curve) with the corresponding degradation of the analytical parameters among runs (inter-day LoD, accuracy, and precision). The reaction time proved to be the most flexible variable, especially when optimizing the procedure for the measurement of large sample batches. Incubation at 55 °C for 60 min with 12.5 mM VCl_3_ yielded the best results with respect to detection limit and precision. A first-tier estimation of the reduction yield in the different conditions was calculated as the ratio of the slopes of the nitrate assay with reference to that of the nitrite assay run in the same optimization or measurement batch. Values in excess of 80% were always obtained (84 ± 4% on 9 batches on different days) in the established method conditions.

#### 2.1.3. Linearity, Detection Limit, Accuracy, and Precision

Separate calibration lines for the measurement of nitrite and combined (nitrite + nitrate) were prepared with each measurement batch.

***Nitrite.*** The Griess reaction, performed on serial dilutions of stock nitrite in the concentrations from 5 to 200 μM, afforded a constant linear relationship of absorbance to concentration throughout the range. Correlation coefficients were always higher than 0.9995 (n = 18) and yielded standard errors of measurement in the range of 0.5 to 1.5 μM. Equation parameters feature a virtually nil intercept (10.1 × 10^−3^ ± 1.0 × 10^−3^) and a constant slope of 4.048 × 10^−3^ ± 0.011 × 10^−3^ (CV 6% on 9 day-to-day batches).

The LoD, calculated from the parameters of the regression line as three times the standard deviation of the intercept, is between 0.5 and 2.5 μM (2.3 ± 0.8; 18 independently measured calibration lines in different days). The accuracy was assessed by re-calculating the concentration of the water-based calibrators on the best-fit regression line. Precision was assessed as the percent coefficient of variation of the re-calculated concentration of the water-based calibrators on the best-fit regression line. The Limit-of-Quantification (LoQ) was estimated as the lowest calibrator for which accuracy is within ±20% of the target, and the precision, expressed as the percent coefficient of variation (CV%), is lower than 20%. In all individual calibration lines associated with the measurement batches, the lowest calibrator at 5 μM was within the stated limits (see Appendix A, Calculation, and data analysis. Nitrite and nitrate concentration).

Results are reported in Table 1.

The typical reference line of Figure 2 was employed to assess the efficiency of nitrate reduction to optimize the reaction conditions.

***Nitrate (total nitrogen oxides; NOx).*** The Griess reaction performed on serial dilutions of stock nitrate in concentrations ranging from 5 to 200 μM, subject to an in-situ reduction with the vanadyl reagent, afforded a constant linear relationship of absorbance to concentration throughout the range, with the correlation coefficients always higher than 0.9995 (n = 18). A typical reference line is illustrated in Figure 3.

Reproducible values of the intercept (124 × 10^−3^ ± 3.83 × 10^−3^) and slope (3.53 × 10^−3^ ± 0.04 × 10^−3^) afforded to achieve a standard error of measurement in the range of 1.6–1.9 μM. The LoD, calculated from the parameters of the regression line as three times the standard deviation of the intercept, is between 1.8 and 3.7 μM (3.1 ± 1.8; 12 independently measured calibration lines in different days).

The accuracy was assessed by re-calculating the concentration of the water-based calibrators on the best-fit regression line. Precision was assessed as the percent coefficient of variation of the re-calculated concentration of the water-based calibrators on the best-fit regression line. The Limit-of-Quantification (LoQ) was estimated as the lowest calibrator for which accuracy is within ±20% of target and precision, expressed as the percent coefficient of variation (CV%) is lower than 20%. In all individual calibration lines associated with the measurement batches, the lowest calibrator at 5 μM was within the stated limits (see Appendix A, Calculation, and data analysis. Nitrite and nitrate concentration.).

Results are reported in Table 2.

This general method is optimized for the measurement in two blood fractions: plasma and red blood cells, each of which entails a specific pre-analytical sample treatment. The main results for each biological matrix are reported separately.

Boundary conditions for method development are given by the available laboratory equipment, which is centered on a plate reader that accommodates 96-well standard plates with a volume of 250 μL per well and on manual sample handling with single- or multichannel pipettors. The main source of uncontrolled absorbance variability is due to the well-to-well irregularity of the sample and reagents volume feeding and possible sample evaporation in the chemical reduction phase.

### 2.2. Measurement in Plasma

#### 2.2.1. Pre-Analytical Treatment of Plasma

The elimination of plasma proteins by precipitation with chilled acetonitrile, followed by centrifugation resulted in the most viable procedure. The optimal volume ratio of plasma:acetonitrile allowing minimum sample dilution was 1:2, *v*/*v*. Using 100 μL of plasma allows the 200-μL sample volume necessary to perform the complete test on a single aliquot.

#### 2.2.2. Nitrite and Nitrate Recovery in Plasma

Due to the difficulty of obtaining nitrate- and nitrite-free plasma in a sufficient volume to prepare method calibrators throughout the several measurement batches, the use of water calibrators prompted the necessity of performing a recovery study to account for the efficiency of the nitrate-to-nitrite reduction step of the method.

Pools of plasma samples with unknown concentrations of nitrite and nitrate were prepared from frozen and thawed leftovers of previous studies and portioned into three aliquots for spiking. One was spiked with nitrate at two levels of approx. 20 μM and 40 μM. The measurement of nitrate in spiked and corresponding unspiked samples afforded the results reported in Table 3. The recovery of added nitrate was essentially quantitative.

#### 2.2.3. Nitrite and Nitrate Levels in Blood Plasma

To assess the representative levels of nitrite and nitrate expected in human plasma, a fitness-for-purpose study of this method examined samples deriving from a small population of healthy volunteers from the laboratory. Reported are the concentration of nitrite (direct measurement with the Griess reaction), total nitrogen oxides (NOx, measured after nitrate reduction and Griess reaction), and nitrate concentration calculated from the difference of Griess-measured total nitrogen oxides (NOx) after reduction and Griess-measured nitrite without reduction. Results are collected in Table 4.

The accuracy of nitrate concentration (last row) depends on the uncertainty of the determinations of nitrite and nitrate, according to the rules for propagation of random errors [22] that results in 1.3 μM and 2.4 μM respectively for NO_2_^−^ and NO_x_, and 2.7 μM (combined uncertainty) for NO_3_^−^. The detailed treatment of this topic is reported in Appendix B and the operative checklist of Appendix A.

### 2.3. Measurement in Red Blood Cells

#### 2.3.1. Pre-Analytical Treatment and Characterization of Red Blood Cells

The measurement in red blood cells becomes physiologically meaningful only when analyte levels can be referred to a well-defined sample volume. Red blood concentrates are thick viscous liquids that are difficult to transfer in an accurately measured volume. The most expedient way to cope with this problem is diluting the concentrate to obtain a hemolysate that can be transferred with higher accuracy. The concentration of hemoglobin in the volume of RBC concentrate taken for the analysis is the auxiliary variable necessary for referring to the concentration of nitrite and nitrate in the aliquot used for the analysis. To refer the measured amount to the original concentration in the RBC, a hemocytometric examination of the original fresh whole blood sample supplies the concentration of hemoglobin and the volume fraction of packed red blood cells (hematocrit) that are necessary for the final calculation.

These operations require an assessment of the uncertainty of the three involved measurements, i.e., the concentration of hemoglobin and the volume fraction of erythrocytes (hematocrit) in the blood sample and the concentration of hemoglobin in the RBC concentrate taken for the analysis.

Uncertainty of the measurement of blood hemoglobin (HGB, g/dL) and hematocrit (HT, as a percent) in the automated blood analyzer were assessed through a quintuplicate measurement of a whole blood sample. Hemoglobin measurement yielded a mean concentration of 15.2 ± 0.8 g/dL (RSE 5.2%) and hematocrit 44.9 ± 2.7% (RSE 5.9%). The resulting concentration of hemoglobin in the red blood cells is calculated as the product of HGB * HT and converted from mass units to molecular units, and yields for the examined blood sample a value of 5.24 ± 0.42 mmol/L (CV 7.9%) red blood cell hemoglobin concentration.

A representative 80-sample batch of red blood cells was used to assess the uncertainty of hemoglobin concentration in the diluted samples prepared for protein precipitation. Concentration was measured by further diluting the samples at 1:50 and comparing them to a standard curve of human hemoglobin. Concentration in the individual diluted samples ranged from 7.0 ± 0.27 (RSE 3.85%) to 65 ± 0.54 (RSE 0.83%) μM. Therefore, hemoglobin concentration in the hemolysates used for nitrite and nitrate measurement was in the range 0.35 ± 0.014 (RSE 3.85%) to 3.24 ± 0.027 (RSE 0.83%) mM, depending on the amount of centrifuged RBC that had been pipetted into the cone during sample fractioning.

A 100-μL sample of this hemolysate, which thus corresponds to a well-defined blood volume, is used for RBC protein precipitation and measurement of nitrite and nitrate in the supernatant.

As reported in the Appendix B (Figure A1), we tested whether (too low) hemoglobin concentration in the hemolysate negatively influences the measurement of low-level nitrite in the samples. We could conclude that, with our present sensitivity, we can measure low μM of nitrite in RBC samples that are (by mere chance) smaller than the 50 μL of concentrated RBC that is standard in blood fractionation.

The elimination of RBC proteins, mostly hemoglobin, is achieved by precipitation with chilled acetonitrile, followed by centrifugation, with the same considerations discussed above for plasma matrix.

#### 2.3.2. Nitrite and Nitrate Recovery in Red Blood Cell Hemolysate

Pools of red blood cell hemolysates with unknown concentrations of nitrite and nitrate were prepared from frozen and thawed red blood cell concentrates and portioned into three aliquots for spiking. One was spiked with nitrite at two levels of approx. 2 μM and 4 μM. Separate aliquots of the same sample were spiked with nitrate at two levels of approx. 20 μM and 40 μM. Measurement of nitrite and nitrate in spiked and corresponding unspiked samples afforded the results reported in Table 5. Recovery was essentially quantitative.

In addition, to verify that acetonitrile-precipitated hemoglobin does not hold a measurable pool of nitrite, as heme- or cysteine-bound NO, the protein pellets from a batch of 24 red blood cell samples were digested with 1 M HCl at 55 °C for 30 min. The absorbance of the resulting solution was assayed at 450 nm to assess the presence of released heme and at 540 nm for its possible interference with the Griess nitrite assay. Only 6/24 samples yielded higher-than-LoD measurements at 450 nm, corresponding to 1.7–3.1 μM Hb. A corresponding interfering absorbance was measured at 540 nm in the range of 3 μM (1.9–3.5 μM) in the absence of any Griess reaction. Performing the assay on the same samples did not increase the absorbance (mean absorbance difference corresponding to −0.5 ± 1.7 μM nitrite), thus demonstrating that no acid-labile Griess-reacting substance is released from the precipitated protein pellet.

#### 2.3.3. Nitrite and Nitrate Levels in Red Blood Cells

To assess the representative levels of nitrite and nitrate expected in human red blood cells, a fitness-for-purpose study of this method examined samples deriving from a small population of healthy volunteers from the laboratory.

The measurements in Table 6 are reported in three successive tiers of data elaboration. First, the “raw” concentration as measured in the diluted hemolysate (μM) is reported. Second, the previous measurement is normalized to a constant concentration of hemoglobin (μmoles/mmol Hb), considering that in the individual samples, the hemoglobin concentration ranges from 1.37 to 1.86 mM. Third, the concentrations of nitrite and nitrate in the RBCs of the individual subjects are reported (μM), considering that the individual subjects have RBC hemoglobin concentrations that range from 5.18 to 5.59 mM.

The biological variability of nitrate and nitrite concentration in RBC resulted higher than in plasma, with two of the six control subjects showing a very high concentration of nitrate. The meaning of this observation needs further confirmation from a larger sample pool.

The accuracy of the nitrate concentration depends on the uncertainty of the determinations of nitrite and nitrate, according to the rules for propagation of random errors. In addition, the concentration of hemoglobin in the diluted hemolysate and the original whole blood sample are used to calculate the concentration in the red blood cell. The hemoglobin concentration in the analytical sample is assessed by spectrophotometry in the plate reader and is subject to its measurement uncertainty. The original whole blood sample is characterized by measuring its hemoglobin concentration and the volume fraction of red blood cells (hematocrit) in standard clinical chemistry equipment, where each measurement is affected by uncertainty. The assessment of extended uncertainty for nitrite and nitrate concentration in red blood cells is reported in Appendix A to demonstrate the employed calculation method.

### 2.4. Pilot Study for the Fitness-for-Purpose Assessment

This method is developed for employing in the frame of physio-pathological studies of human subjects in extreme environment conditions [5] and patients with several chronic and acute conditions, such as those who need assisted ventilation [23]. The assessment aims at optimizing study conditions, sample management, and measurement methods in large-batch studies where sample allotment is critical due to the difficulty of blood sampling and to the scarce volume available.

To show the ability of the combined nitrite-nitrate measurement to discriminate values that change over the physiological range, two different situations are shown as examples. In Figure 4, blood was sampled from a scuba diver before, during (underwater), and after training in a land-based deep-water facility Y-40 pool [5]. The sharp rise of the nitrate concentration 10 min after the beginning of the dive (point 2, red diamond) shows the known vasodilatory response of the organism to counteract the squeezing effect of a two-atmosphere overpressure on the body. During this experiment, red blood cells could not be recovered. In Figure 5, nitrites and nitrates were assayed in RBC of a subject who dwelt for six months in the Antarctica Research Station, and six blood samples were withdrawn from the subject before the journey and at 3, 7, 20, and 90 days after arrival at the Station and immediately before leaving. In this experiment, the body seems to counteract the extreme conditions of Antarctica after 3 days from the arrival, while after many months, before leaving, it seems to have adapted. For the subjects of this study, plasma could not be assigned for this measurement.

Both graphs display the extended uncertainty of the measurements as vertical bars accompanying each data point. Its value in plasma is approx. 0.6 μM for nitrate and 2.3 μM for total nitrogen oxides (combined for nitrate is 2.4 μM). In red blood cells, that for nitrite is approx. 0.9 μmoles/mmole hemoglobin, 2.3 μmoles/mmole hemoglobin for total nitrogen oxides, and 2.4 μmoles/mmole hemoglobin for nitrate. The achieved total measurement uncertainty is at all sufficient to highlight differences in the time course of nitrate concentrations. Doubling of nitrite concentrations with basal levels as low as 2 μM can be discriminated, too, due to the sufficiently low uncertainty of the measurement.

To display the range of measured values in plasma and in the red blood cells of subjects in the performed studies, the histograms of Figure 6 and Figure 7 show the levels of nitrite and nitrate in subjects that derive from studies performed in extreme environments. Results are displayed as ordered by increasing total nitrite + nitrate concentration without reference to the specific studies they belong to.

Figure 6 shows levels of nitrite (blue bars) and nitrate (red bars) in one hundred plasma samples of scuba divers who performed training at four different underwater depths and were sampled out of the water, before each trial and at two times after the exercise.

Figure 7 shows levels of nitrite (blue bars) and nitrate (red bars) in eighty red blood samples of subjects who participated in two studies under the extreme conditions of breath-hold diving in the open sea (8 subjects; n = 32 samples) and of dwelling for six months in the Concordia Antarctica station (9 subjects; n = 48 samples).

For a further, 8-subject, 24-sample pilot study, plasma and red blood cell collection were performed on-site at the land-based deep pool diving facility, and the measurement of blood hemoglobin and hematocrit could be performed within 24 h, at the laboratory. This sample collection and analysis procedure is, at present, the fastest and most complete possible and affords concentrations of nitrite and nitrate both in plasma and in the red blood cells. To appreciate the increase in information that this procedure affords, results are displayed in Table 7.

## 3. Discussion

Although several published procedures already exist to measure nitrite and nitrate in biological samples, especially in plasma or serum, for the aims of biochemical and clinical research, several parameters need to be in control to obtain quality measurements, on which biological hypotheses and explanations are grounded. In our application, sample economy is also at a premium because blood samples often derive from studies of humans at remote locations.

One main aim of the developed method is the lowest-possible sample consumption compatible with measurement accuracy and precision. While in principle, a 50-μL sample volume may be sufficient in a 1:2 scaled-down procedure performed in a commercially available low-volume 96-well sample plate, the higher uncertainty of well-to-well volume transfer with manual equipment may reflect an unnecessarily higher standard error, especially for determinations at the lower concentration.

Literature suggests three methods for the reduction of nitrate to nitrite: (a) chemical reduction with Cadmium metal under heterogeneous reaction conditions (Jones method); (b) chemical reduction with the vanadyl ion with the use of Vanadium(III) chloride (VCl_3_); and (c) the use of nitrate reductase and NADH.

The use of Cadmium granules in packed columns for the Jones method is widely employed, especially in automatic water analyzers [18]. However, the method employs a carcinogenic reagent, for which a risk assessment procedure should be performed, and use is not authorized when alternatives exist. In addition, in our application, the millimeter size of available granules lends poor to the manipulation of small-volume plasma samples. Recovery and disposal of spent granules is itself a tedious and hazardous procedure. Recovery for re-use is also a procedure that may entail the chance of sample cross-contamination and additional risk for the laboratory operators. For this reason, this procedure was sidelined from the planning phase.

The use of a nitrate reductase preparation was examined as a possible alternative due to its specificity and the favorable risk assessment profile. However, the cost of the enzyme and its cofactor NADPH, the limited stability over time of the reagents, and the availability of a viable chemical procedure discouraged the development of this less expedient alternative.

During method development, the reduction of nitrate to nitrite with vanadyl chloride proved a critical step, for which a cautious control of reaction conditions in the biological extract resulted as essential to achieve the required analytical performance. The optimized conditions, with 60 min incubation at 55 °C in a lid-covered 96-well plate for the complete development of the color reaction, allows completing a two-plate, 80-sample batch analysis of plasma samples in close to three hours.

Analytical results presented in this fitness-for-purpose assessment were from an Antarctica remote exploration facility (the Concordia Research Station, at http://www.concordiastation.aq/home-1/; last accessed on 27 July 2021) [20] and from subjects in extreme conditions by blood sampling from divers at sea and in a land-based deep-water training facility (the Italian Y-40 “The Deep Joy”, at https://www.y-40.com/en/; last accessed on 27 July 2021) [21]. Both are examples of contemporary physiological research aimed at studying the response of the organism to physiological extremes of temperature cycling, external pressure, and exposure to varying concentrations of inhaled oxygen concentration.

Red blood cells are becoming of increasing interest as the subject of physiological studies because they are, per se, isolated but communicating cells, simple but viable biological systems that respond to the physiological and environmental conditions of the whole organism. In addition, most studies measure biomarkers in plasma or serum, while red blood cells are usually not considered. Thus, this blood fraction would not compete for sample volume allotment with other measurements where the use of plasma or serum is mandatory, such as the soluble immune system proteins [24], coagulation factors, circulating hormones, and neuro-endocrine parameters [25] and the plasma metabolome [26].

The management of this biological compartment for measurements, however, shows its own difficulties due to the necessity to refer measured amounts of analytes to physiologically meaningful parameters of red blood cell volume in order to achieve representative and comparable results. One main difficulty is that both concentrated RBC of centrifuged whole blood and frozen and thawed red blood cell concentrates are difficult to transfer and measure with sufficient accuracy. In addition, frozen and thawed blood and RBC concentrates hemolyze, and the levels of several sensitive biomarkers, such as antioxidants, would change and yield physiologically meaningless results. The complete measurement in red blood cells thus includes that of the hemoglobin concentration in the hemolysate, nitrite, and nitrate in three separate plates, each with its calibration curve.

In addition, field studies outside research hospital facilities require minimal blood treatment immediately after withdrawal, due to the scarcity of time for processing, the possible lack of trained technicians, and the inadequacy of physical facilities for safe blood fractionation.

At the time of analysis, RBC should be diluted to overcome sample volume heterogeneity during the transfer for analytical sample preparation, and the analytical results are referred to the amount of hemoglobin in the sample. This parameter is a strong proxy to the actual amount of considered sample and can be referred to actual concentration in RBC by normalization of measured hemoglobin concentration to that in the original blood sample or standard values of hematocrit and blood hemoglobin concentration.

## 4. Materials and Methods

The checklist document (Appendix A) reports the full method details, including safety notes and calculation examples.

### 4.1. Equipment

#### 4.1.1. Reagents

Sodium nitrite and sodium nitrate of analytical purity were of standard availability. The Griess reagents sulphanylamide (SA) and *N-*(1-naphthyl)ethylenediamine (NED) were obtained from Sigma-Aldrich (Milano, Italy). Acetonitrile for protein precipitation was of chromatographic purity (Merck, Milano, Italy) and used as received. Deionized water was supplied from a MilliQ department facility.

#### 4.1.2. Glass and Plasticware

Standard class-A glass volumetric flasks of 25 mL nominal volume were used to weight and dilute the initial stock solutions of Sodium nitrite and Sodium nitrate.

Successive dilutions were performed and stored at +4 °C, in stoppered standard polypropylene tubes.

All pre-analytical manipulations of biological samples were performed in Eppendorf-type plasticware of a standard laboratory availability, brands of which were specially selected for rugged use in field sampling and quick fractionation of whole blood.

Precipitation of plasma proteins was performed in 1.5 mL centrifuge tubes.

The spectrophotometric measurements were obtained in 96-well polystyrene plates with a well volume of approx. 250 μL and a well-fitting lid.

#### 4.1.3. Measurement Equipment

A Medonic M20 hemocromocytometer (Boule, Spånga, Sweden) was employed to measure hematological parameters in the freshly withdrawn blood samples.

A Micro Star 17 minifuge (VWR, Radnor, PA, USA) was employed for protein precipitate separation.

An FD-23 laboratory oven (Binder, Tuttlingen, Germany) was used for plate heating.

A MicroPette 8-channel multiple pipettor (Dlab Scientific, Locust St, Welland, ON, Canada) with disposable tips was used for dispensing liquids in the derivatization step.

A Sunrise microplate reader (TECAN, Salzburg, Austria) with filters was employed for spectrophotometric measurements. It operated at 450 nm (filter bandpass 8.5–16 nm) for hemoglobin measurement and at 540 nm (550 nm filter bandpass 10–14 nm) for the measurement of the Griess reaction product.

#### 4.1.4. Reagent and Standard Solutions

Standard solutions of nitrite and nitrate are prepared in a large concentrated batch to allow weighting a sufficiently large quantity of the salts (1 mmole) to ensure precision within ±1%.

The hemoglobin standard solution is prepared freshly before the measurement.

*Hemoglobin.* An approx. 2.5 mg sample is weighted in a 2-mL Eppendorf tube and diluted to a final concentration of 40 μM in deionized water. Appropriate volumes are serially diluted to 20, 10, and 5 μM (e.g., 0.5 mL solution + 0.5 mL water) with deionized water for spectrophotometric measurement.

*Sodium nitrite.* An approx. 70 mg sample is weighted in a 25-mL volumetric flask, and appropriate volumes are serially diluted to 200, 100, 50, 20, 10 and 5 μM with a mixture of deionized water and acetonitrile in a 1:2 volume ratio.

*Sodium nitrate.* An approx. 90 mg sample is weighted in a 25-mL volumetric flask, and appropriate volumes are serially diluted to 200, 100, 50, 20, 10, and 5 μM with a mixture of deionized water and acetonitrile in a 1:2 volume ratio.

The reagents are prepared in appropriate amounts shortly before each analytical run.

*Vanadyl chloride.* For the batch of 96 samples, approx. 40 mg VCl_3_ (MW 157.30) is weighted in a 10-mL stoppered plastic tube and diluted to a final concentration of 25 mM in 1 M hydrochloric acid.

*Sulphanylamide (Griess reagent A).* For the batch of 96 samples, approx. 150 mg (MW 172.20) is weighted in a 10-mL stoppered plastic tube and diluted to a final concentration of 87 mM in 2M hydrochloric acid.

*N-(1-naphthyl)ethylenediamine (Griess reagent B).* For the batch of 96 samples, approx. 10 mg (MW 259.18) is weighted in a 10-mL stoppered plastic tube and diluted to a final concentration of 3.9 mM in 5% *v/v* phosphoric acid.

*Combined Griess reagent.* For the batch of 96 samples, 10 mL of reagent A and 10 mL of reagent B are combined shortly before sample preparation.

All excess reagents, spent plate content and excess biological samples and extracts are disposed of after use in a safety bag, according to national regulations.

### 4.2. Sample Preparation and Measurement

Auxiliary measurements include a hemocytometric characterization of each fresh blood sample that includes total hemoglobin concentration (mg/dL), hematocrit fraction (%) and red blood cell number (number/μL) and volume (fL; 1 fL = 10^−12^ mL). For samples obtained fresh at the laboratory, a 150-μL sample of whole blood is dispensed in a 0.5-mL cone and submitted to the instrument for the measurement. For field studies, this measurement is performed as early as possible on whole blood stored and transported in a +4 °C chain.

The procedure is optimized to measure samples with minimal use of disposable plasticware by performing all spectrophotometric measurements in as small a number as possible of a 96-well plate. A small study number of 13 samples can be measured in a single 96-well plate, along with duplicate standard curves. A complete 24-sample study can be performed in two plates, one for plasma and one for the red blood cells, the latter including hemolysate hemoglobin measurement.

A complete checklist procedure is reported as Appendix A, with separate sections for measurement in plasma/plasma and red blood cells since the latter also entails hemocytometric and hemoglobin concentration measurement as an essential auxiliary result. Briefly, the following steps are followed.

#### 4.2.1. Protein Precipitation

***Procedure for plasma***. From each thawed and mixed plasma sample, 100 μL were pipetted into a 1.5 mL Eppendorf-type centrifuge tube. To each tube, 200 μL of chilled (−20 °C) acetonitrile are added, and the closed tube is immediately vortexed for 5 sec, then each lot of 6 samples centrifuged at 12,000 rpm for 10 min. Two 100 μL subsamples are withdrawn from each for separate nitrite and nitrate measurement.

***Procedure for red blood cells***. To each thawed sample tube containing approx. 50 μL of separated RBCs, 100 μL of chilled sterile water were added, the tube stoppered and mixed by inversion to completely hemolyze and resuspend the contents. A 10 μL subsample was withdrawn and diluted at 1:50 for immediate measurement of hemoglobin concentration (see below).

A 100 μL subsample was dispensed in a 1.5 mL Eppendorf tube, followed by 200 μL of chilled acetonitrile. Each lot of 6 stoppered tubes were immediately centrifuged at 12,000 rpm for 10 min, and two 100 μL subsamples were withdrawn from each for separate nitrite and nitrate measurement.

#### 4.2.2. Sample Plate Preparation and Measurement

According to the number of measured samples, the determinations of nitrite, nitrate, and hemoglobin are performed in the same plate with successive spectrophotometric readings, or one plate per measurement is prepared, with samples in matching positions.

Fresh reagent batches were prepared from the solids and used immediately after.

In a 96-well plate, for each well, the following volumes of sample and reagents are dispensed.
(a)In total, 100 μL of the sample (from the protein precipitation steps above) or of the water-based calibration solution is pipetted.(b)A total of 50 μL of the combined Griess reagent mixture is added with the 8-channel multiple pipettes.(c)Then, for the measurement of nitrite + nitrate, 100 μL of Vanadium(III) chloride (25 mM in 1 M HCl) solution is added with the 8-channel multiple pipettes. For the measurement of nitrite alone, 100 μL of 1 M HCl is added in place of VCl_3_ solution with the 8-channel multiple pipettes.

For the measurement of nitrite alone, the plate is allowed to stand at room temperature for 15 min before insertion in the plate reader for absorbance measurement.

For the measurement of nitrate, the plate is incubated at 55 °C for 60 min, covered with its lid, then allowed to cool at room temperature for 5 min. The plate is then inserted into the reader for the spectrophotometric measurement.

Absorbance measurements from the results collection sheet are input into the calculation spreadsheet.

#### 4.2.3. Standard Curves Preparation

A six-level standard curve at 0–5–10–20–50–100–200 μM is analyzed in duplicate for each lot of samples analyzed for nitrite and nitrate.

For the measurement of hemoglobin, a four-level standard curve at 40–20–10–5 μM is analyzed in duplicate.

#### 4.2.4. Sample Plate Reading

The plate for the measurement of nitrite and nitrate is read at 540 nm (550 nm, 10–14 nm bandpass filter).

The plate for the measurement of hemoglobin concentration is read at 450 nm (450 nm, 8.5–16 nm bandpass filter).

### 4.3. Calculation

#### 4.3.1. Data Input

Measurements from the different instruments are downloaded to a custom spreadsheet with pre-coded information on sample identity. Ordered data is transferred to custom spreadsheets for analytical calculations and results in normalization.

All calculations were performed with custom spreadsheets of the laboratory, validated against the calculation examples in the 1984 edition of Miller & Miller’s *Statistics for Analytical Chemists* [22]. The Microsoft Excel spreadsheet was used for data plotting. The checklist document (Appendix A) reports the full calculation details.

#### 4.3.2. Calculation of Analyte Concentration

Standard curves are elaborated with least-square straight-line fitting. One step of iteration is allowed to account for irregularities in liquid dispensing from the multichannel pipette and from using wells rather than fixed-length cuvettes for spectrophotometric reading.

Absorbance data from the coded samples are transferred to the appropriate section of the spreadsheet, where the calculation of the concentrations and associated standard error of determination is performed. Speciation of nitrogen oxides is performed by subtracting the concentration of nitrite from that of the sum of nitrite and nitrate.

Measurements in red blood cells are normalized per millimole of hemoglobin. Further processing yields actual concentration in the red blood cell, with the use of the values of total blood hemoglobin and of hematocrit fraction of the individual blood sample.

#### 4.3.3. Calculation of the Extended Error of Determination

The extended error of determination is calculated from the error associated with each determination [22].

For the determination of extended uncertainty of nitrate concentration in the sample from the separate measurement of nitrite and total nitrogen oxides (NO_x_), Equation (1) is used, where the Standard Deviation (SD) of the measurements are employed.
SD(NO_3_) = (SD(NO_2_)^2^ + SD(NO_x_)^2^)^1/2^(1)

For the determination of extended uncertainty of concentrations normalized to hemolysate hemoglobin concentration, Equation (2) is used, where the Coefficient-of-Variation (100*SD%) of the measurements are employed.
CV(NO_3_)/Hb = (CV(NO_2_)^2^ + CV(Hb)^2^)^1/2^(2)

A two-tail *t*-test was used to verify whether two measurements (e.g., sequential in time, each with calculated uncertainty) were statistically different.

## 5. Conclusions

This method is currently employed in our laboratory for the measurement of nitrite and nitrate in plasma and red blood cells of subjects who participate in population and clinical studies. Current areas of interest include the effect of the NO pool on the biological response of the vascular system to extreme cold, hypoxia, extreme variations of environmental pressure, and pathological conditions that require assisted ventilation. The low uncertainty of this method allows discriminating meaningful variations of as little as 2 μM in the plasma and red blood cell concentration of nitrite. Its dynamic range over at least two orders of magnitude, up to more than 200 μM, allowed pinpointing the vasodilatory response to a 3-atmosphere overpressure on the body of a diving swimmer.

## Figures and Tables

**Figure 1 molecules-26-04569-f001:**
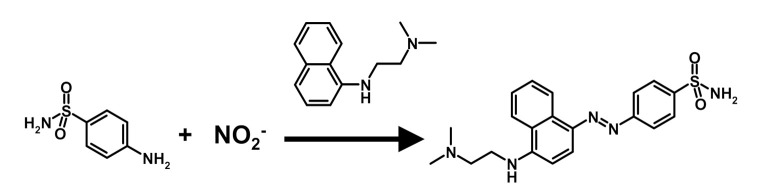
The reaction scheme of the Griess diazotization reaction.

**Figure 2 molecules-26-04569-f002:**
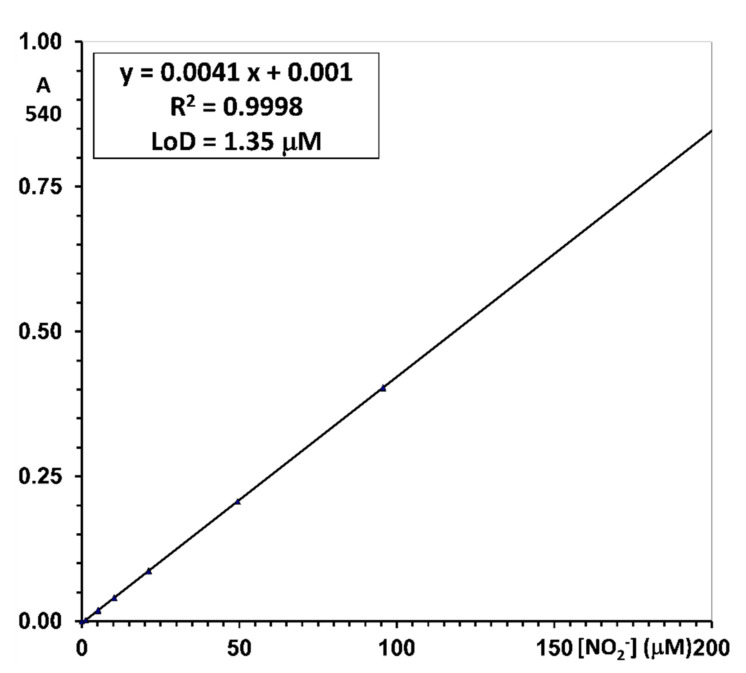
An example of a daily standard curve for nitrite measurement with the reported equation, goodness-of-fit, and Lower Limit of Detection (LLoD). Confidence limits of ±0.9 μM are highlighted by the dashed lines. Analytical conditions are described in the Materials and Methods section.

**Figure 3 molecules-26-04569-f003:**
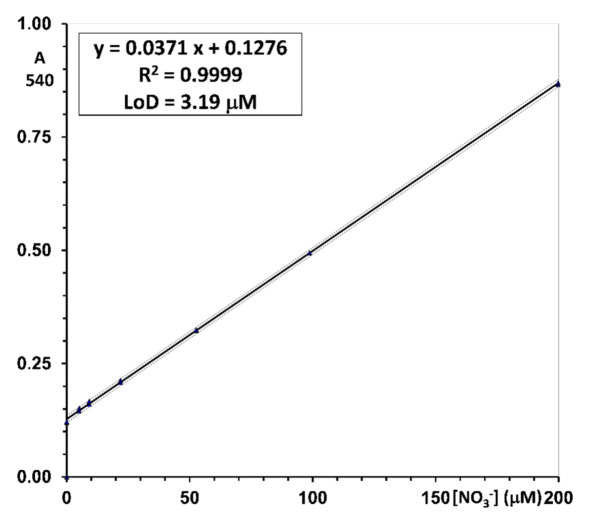
An example of a daily standard curve for nitrate measurement with the reported equation, goodness-of-fit, and Lower Limit of Detection (LoD). Confidence limits of ±1.6 μM are highlighted by the dashed lines. Analytical conditions are described in the Materials and Methods section.

**Figure 4 molecules-26-04569-f004:**
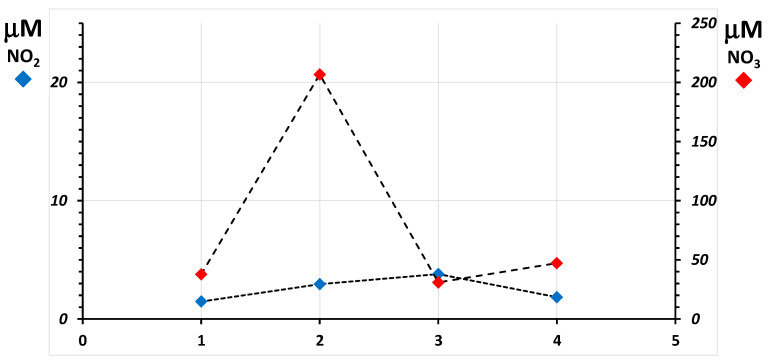
Line plot of the concentration of nitrite (blue diamonds) and nitrate (red diamonds) in plasma measured in one scuba diver before, during, and after training in a land-based deep-water facility. Blood samples were obtained from the subject before the exercise (**1**), at −20 m depth, 10 min after the beginning of the dive (**2**), 15 min after surfacing (**3**), and 45 min after surfacing (**4**). The plot shows, for each measurement, the associated combined uncertainty calculated from those of nitrate and of total nitrogen oxides in the same sample.

**Figure 5 molecules-26-04569-f005:**
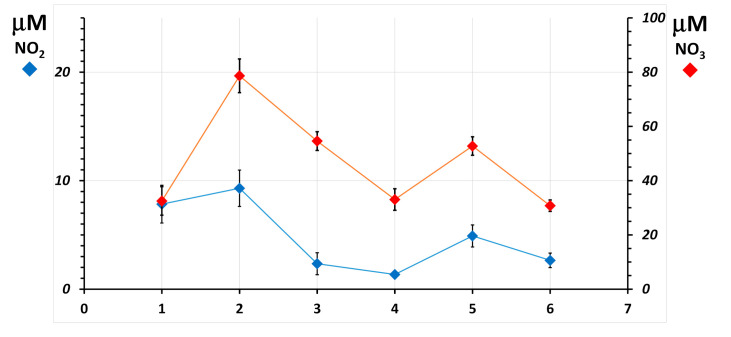
Line plot of the concentration of nitrite (blue diamonds) and nitrate (red diamonds) in RBC, normalized for hemoglobin concentration as μmoles per millimole hemoglobin measured in one subject of the Concordia Arctic Research Station 2015 study. Blood samples were obtained from the subject before the journey (**1**) and at 3 (**2**), 7 (**3**), 20 (**4**), and 90 (**5**) days after arrival at the Station and immediately before leaving (**6**). The plot shows, for each measurement, the associated combined uncertainty calculated from those of the individual measurements of hemoglobin, nitrate, and total nitrogen oxides in the same sample.

**Figure 6 molecules-26-04569-f006:**
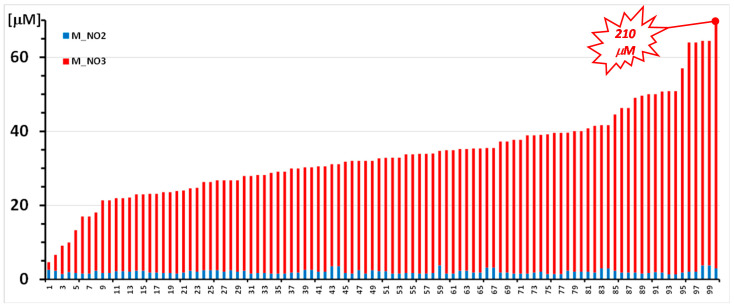
Nitrite (blue bar) and nitrate (red bar) concentrations were measured in 100 plasma samples from 11 anonymized scuba divers who participated in a training session at three progressively deeper water levels in a land facility. The concentration of both analytes is reported as μmoles/L. No sample yielded nitrite concentration < LoD. Sample 100 reached a nitrate concentration of 210 μM (sample 2 of Figure 4).

**Figure 7 molecules-26-04569-f007:**
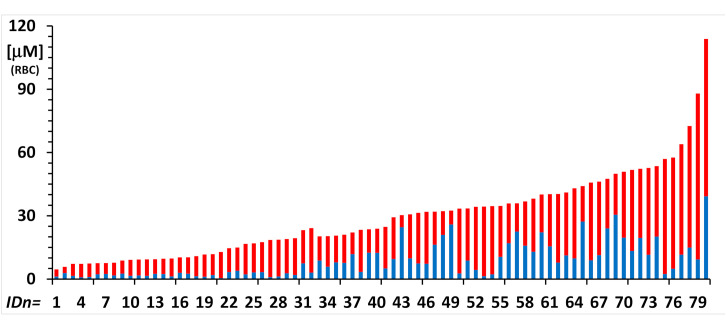
Nitrite (blue bar) and nitrate (red bar) concentrations were measured in 80 RBC samples from anonymized subjects who participated in two studies under the extreme conditions of breath-hold diving (8 subjects; n = 32 samples) and of dwelling for six months in the Concordia Antarctica station (9 subjects; n = 48 samples).

**Table 1 molecules-26-04569-t001:** The accuracy and precision of the nitrite measurement ^1^.

Nominal (μM)	Recalculated (μM)	SD (μM)	Accuracy ± %	Precision CV%
0	0.2	0.6		
5	3.8	1.2	−24%	31%
10	11.3	2.1	13%	19%
20	21.9	3.0	10%	14%
50	54.4	5.1	9%	9%
100	101.5	2.1	2%	2%
150 ^2^	149.9	1.2	0%	1%
200	198.3	2.4	−1%	1%

^1^ Reported results are the mean of individual ones based on 18 independently prepared measurement batches. ^2^ Only 5 batches.

**Table 2 molecules-26-04569-t002:** The accuracy and precision of the nitrate measurement ^1^.

Nominal (μM)	Recalculated (μM)	SD (μM)	Accuracy ± %	Precision CV%
0	−2.2	5.5		
5	5.5	2.6	−11%	48%
10	9.5	1.1	3%	11%
20	19.9	4.7	−22.0%	24%
50	51.3	2.1	8.2%	4%
100	101.2	2.8	3.4%	3%
150 ^2^	149.2	6.2	−0.5%	4%
200	199.2	1.6	−0.7%	1%

^1^ Reported results are the mean of individual ones based on 18 independently prepared measurement batches. ^2^ Only 5 batches.

**Table 3 molecules-26-04569-t003:** The recovery of nitrite and nitrate in human plasma.

Sample	Nominal (μM) ^1^	Recalculated (μM) ^1^	Accuracy ± %	Precision CV%	Rec %
Plasma (not fortified) ^1^	1.3			1.1%	
NO_2_^−^ -fortified. (10.3 μM) ^1^	11.6	11.5	99.1%	1.3%	99.4%
NO_2_^−^ -fortified. (20.6 μM) ^1^	21.9	22.0	100.5%	1.4%	100.5%
Plasma (not fortified) ^1^	40.4			6%	
NO_3_^−^ -fortified (21.0 μM) ^1^	61.4	61.2	99.7%	2%	97.3%
NO_3_^−^ -fortified (42.0 μM) ^1^	82.4	87.4	106.0%	3%	104.6%

^1^ Results as the mean of two determinations.

**Table 4 molecules-26-04569-t004:** Nitrite and nitrate concentration in plasma of volunteer subjects.

Case Studies	Subj. ID	NO_2_^− 1^	NO_x_ ^1^	NO_3_^− 1^
Control Subjects (n = 6)	AB79M	1.9 ± 1.3	34.8 ± 2.4	32.9 ± 2.7
	FP89F	2.1 ± 1.3	44.2 ± 2.4	42.1 ± 2.7
	FR60M	2.1 ± 1.3	32.0 ± 2.4	29.9 ± 2.7
	GP97F	1.9 ± 1.3	28.1 ± 2.4	26.2 ± 2.7
	GS82M	2.5 ± 1.3	29.5 ± 2.4	27.0 ± 2.7
	MS51M	2.3 ± 1.3	40.6 ± 2.4	38.3 ± 2.7
	Mean	2.1	34.9	32.7
	S.D.	0.3	6.4	6.4
	CV%	14.2	18.3	19.5

^1^ results in μmole/L (µM).

**Table 5 molecules-26-04569-t005:** The recovery of nitrite and nitrate in human red blood cells hemolysate.

Sample	Nominal (μM) ^1^	Recalculated (μM) ^1^	Accuracy ± %	Precision CV%	Rec %
RBC hemolysate (not fortified) ^1^	2.7			0.1%	
NO_2_^−^ -fortified (2.0 μM) ^1^	4.7	4.0	85.2%	0.1%	84.5%
NO_2_^−^ -fortified (4.1 μM) ^1^	6.8	6.7	98.1%	2.3%	99.3%
RBC hemolysate (not fortified) ^1^	51.3			0.4%	
NO_3_^−^ -fortified (20.8 μM) ^1^	72.1	70.2	99.2%	1.4%	97.3%
NO_3_^−^ -fortified (41.6 μM) ^1^	92.9	97.2	105.1%	1.4%	104.6%

^1^ Results as the mean of two determinations in μmole/L (μM) on measured hemolysate.

**Table 6 molecules-26-04569-t006:** Nitrite and nitrate concentrations in red blood cell hemolysates of the subjects used for the method performance study.

Case Studies	Subj. ID				μM in Hemol ^1^			μM/ mM Hb ^2^			μM in RBC ^1^		
		Hb ^3^	Ht% ^4^	Hb ^5^	NO_2_^−^	NOx	NO_3_^−^	NO_2_^−^	NOx	NO_3_^−^	NO_2_^−^	NOx	NO_3_^−^
Control Subjects (n = 6)	AB79M	17.2	48.4	1.77	5.3	30.4	25.1	3.0	17.2	14.2	16.6	95.0	78.4
	FP89F	14.1	40.8	1.86	7.5	29.4	21.9	4.1	15.8	11.8	21.8	85.0	63.2
	FR60M	13.9	41.7	1.74	6.7	35.5	28.8	3.9	20.8	16.9	20.1	107.6	87.5
	GP97F	12.7	35.3	1.74	4.6	20.9	16.3	2.6	12.1	9.4	14.7	67.4	52.6
	GS82M	15.6	46.9	1.37	10.0	72.0	62.0	7.3	52.6	45.2	38.3	274.9	236.6
	MS51M	16.6	47.6	1.65	5.1	19.0	13.9	3.1	11.5	8.4	16.7	62.5	45.8
	Mean	15.0	43.5	1.7	6.5	34.5	28.0	4.0	21.7	17.7	21.4	115.4	94.0
	S.D.	1.7	5.1	0.1	2.0	19.4	17.5	1.7	15.5	13.9	8.7	79.9	71.6
	RSE ^6^										40.6	69.2	76.1

^1^ Nitrite (direct measurement with the Griess reaction; μmole/L, μM); total nitrogen oxides (NOx, measured after nitrate reduction and Griess reaction; μmole/L, μM); nitrate, calculated from the difference of Griess-measured total nitrogen oxides (NOx) and nitrite; μmole/L, μM). ^2^ results in μmole/mmole Hb. ^3^ results in millimole/L from hemochromocytometer. ^4^ results in percent from hemochromocytometer. ^5^ results in millimole/L from plate reader. ^6^ Relative Standard Error as 10^2^ * S.D./Mean.

**Table 7 molecules-26-04569-t007:** Nitrite and nitrate concentrations in plasma and red blood cells of scuba divers.

Case studies	Subj. ID	NO_2_^−^	NO_x_	NO_2_^−^	NO_3_^−^
		Plasma (µM)		RBC (µM)	
Scuba divers at a training facility (Y-40, Italy) (n = 8; range of 3 samples)	S12	<LoD–2.3	37.5–45.6	13.1–42.4	41.3–107.1
	S13	<LoD	36.4–47.5	15.2–25.4	55.7–77.5
	S14	<LoD	35.4–53.4	11.2–64.5	58.1–116.1
	S15	1.9–2.3	74.5–77.3	16.1–29.8	73.5–96.4
	S16	<LoD–1.9	34.8–36.7	17.5–40.4	80.7–106.7
	S17	<LoD–4.2	47.5–58.4	32.7–50.3	46.3–68.2
	S18	<LoD	45.4–55.5	25.2–47.8	69.5–123.8
	S19	<LoD–2.3	48.3–58.1	31.9–58.5	80.6–102.2

LoD(NO_2_^−^) = 1.1 µM; LoD(NO_x_) = 3.5 µM.

## Data Availability

Data used to prepare this article is not publicly available as part of this work. The employed spreadsheets with example calculations are available from the Authors upon reasonable request.

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
