# Peer review of "High-Throughput Griess Assay of Nitrite and Nitrate in Plasma and Red Blood Cells for Human Physiology Studies under Extreme Conditions"

_molecules, 2021, doi:10.3390/molecules26154569_

Round 1

Reviewer 1 Report

The paper is well written and gives a good overview of the issue. Although it is a lengthy paper, the thoroughness of it justifies the length.

Minor points to consider in subsequent versions:

  1. Lower-Limit-of-Detection (LLoD) should be replaced by Limit of Detection (LOD)
  2. The commas in Figures 2; 3 and Tables 1; 2; 5 should be replaced by dots

Author Response

Dear Reviewer 1,

Thank you for your appreciation and the meticulous reading. We have brought some modifications to the original version that consider yours, along the other Reviewers’ comments: 

  1. LLoD has been replaced as suggested
  2. The commas in Figures 2; 3 and Tables 1; 2; 5 has been replaced by dots

Best regards

FMR, on behalf of all Authors

Reviewer 2 Report

The work is of fundamental importance for understanding the regulatory role of nitric oxide in mammals, despite the fact that the subject of study is blood fractions deriving subjects who dwelt in an Antarctica scientific station and on breath-holding and scuba divers who training performed at sea and in a land-based deep pool facility.

Many, many years ago, Griess proposed a reaction for measuring nitrite, which the authors adapted to measure the level of nitrite and nitrate in blood and erythrocyte samples. The proposed development has high sensitivity and, at the same time, high reproducibility, which is important for large-scale measurements.

Preliminary preparation of samples for the removal of plasma proteins by their precipitation with acetonitrile also provides comparable data.

The undoubted advantages of the work include the method used in the work for converting nitrate into nitrites, based on vanadyl ion, ensuring its complete recovery.

Also, for the first time, the authors obtained important data on the content of nitrites / nitrates in erythrocytes and compared them with their content in serum.

In my opinion, the article is structured logically correctly and the research results are clearly stated. Therefore, the introduction into the text of any additional comments is not required, so as not to affect the quality of the work. The article is recommended for publication.

The authors did not aim to explain the sources of nitrites / nitrates in the body, much less to assess the level of nitrosothiols (RSNO) or of dinitrosyl iron complexes (DNIC ). The work of Cialoni D, Brizzolari A, Samaja M,Bosco G, Paganini M, Pieri M,Lancellotti V and Marroni A (2021) Nitric Oxide and Oxidative Stress Changes at Depth in Breath-Hold Diving. Front. Physiol. 11:609642.doi: 10.3389/fphys.2020.609642. is partially devoted to this problem.

Nevertheless, I would like to hear the authors' opinion on the possible effect of RSNO and DNIC on nitrite / nitrate levels.

Was the control of erythrocyte hemolysis carried out under a microscope?

Table 6 presents data from patients with high nitrate levels. Maybe they are smokers?

In this work, deionized water was used to prepare solutions. What is the pH value of this water? I assume that it has a value lower than pH = 7.0 and nitrite is protonated during the preparation of standard solutions. In this case, this explains the scatter of data on nitrite precisely at its low concentrations in Table 1.

Author Response

Dear Reviewer,

thank you for the kind words of appreciation. Here below, please find our answer to your raised points (see attachment). We have brought some modifications to the original version that consider yours, along the other Reviewers’ comments.

Best regards

FMR, on behalf of all Authors

The authors did not aim to explain the sources of nitrites / nitrates in the body, much less to assess the level of nitrosothiols (RSNO) or of dinitrosyl iron complexes (DNIC).

The work of Cialoni D, Brizzolari A, Samaja M,Bosco G, Paganini M, Pieri M,Lancellotti V and Marroni A (2021) Nitric Oxide and Oxidative Stress Changes at Depth in Breath-Hold Diving. Front. Physiol. 11:609642.doi: 10.3389/fphys.2020.609642. is partially devoted to this problem.

Nevertheless, I would like to hear the authors' opinion on the possible effect of RSNO and DNIC on nitrite / nitrate levels.

This is a very difficult issue that deserves a basic approach. It is well known that the only reaction that gives rise to NO in the body is the one that uses arginine as substrate, which can be catalyzed by any of three isoforms of NOS, each with its own Ca++ sensitivity and KM values depending on the cell/tissue compartment where that specific NOS resides. The most relevant NOS here is the endothelial cell NOS. The large majority of produced NO is stored as relatively inert (thus monitorable, see the present study) nitrite/nitrate, with a smaller amount reversibly bound to hemoglobin. In our opinion, the gross amount of nitrite/nitrate in blood greatly exceeds that or RSNO and DNIC by some orders of magnitude. This might indicate that RSNO and DNIC depend on nitrite/nitrate and not viceversa. However, the laws that govern this transformation are yet to be understood. As this is highly speculative, we prefer not to include it in the text.

Was the control of erythrocyte hemolysis carried out under a microscope?

No, because this procedure comes from a very robust protocol generally adopted for hemoglobin adduct measurement. Should any still closed RBC remain, complete hemolysis will take place at the step of protein precipitation.

Table 6 presents data from patients with high nitrate levels. Maybe they are smokers?

All persons of Table 6 are current no-smokers. Divers, too, are non-smokers.

We are not aware of the smoking status of the Antarctica researchers, but likely smoking at the Base was forbidden under safety grounds.

In this work, deionized water was used to prepare solutions. What is the pH value of this water? I assume that it has a value lower than pH = 7.0 and nitrite is protonated during the preparation of standard solutions. In this case, this explains the scatter of data on nitrite precisely at its low concentrations in Table 1.

Submission Date 22 June 2021

Date of this review 07 Jul 2021 22:02:40

The reduction and diazo-copulation reactions are both performed in the necessary strongly acidic medium. We would think unlikely that the pH of water in the water-acetonitrile mixture plays a role.

The between-lots small variability of slope and intercept likely accounts for that of the lower concentration calibrators.

Reviewer 3 Report

1) The hemoglobin concentration was measured spectrophotometrically at 450 nm. Why was this particular wavelength chosen? The hemoglobin Soret band is at ca. 410 nm, followed by Q bands in the 500-600 nm region characteristic for various hemoglobin species (OxyHb, metHb, HbCO, etc.).

2) Nitrate levels were assessed indirectly by the Griess assay proceeded by the reduction with vanadium (added in excess).  The information on the achieved reduction efficiency would be valuable. Also, what hypothesis test was used to determine whether the difference between the two means is significant?

3) The statistical analysis of the presented data is inconsistent and incorrect, e.g., the accuracy is given as a percent accuracy and percent recovery (Table 2 and Table 3). For the same set of data, the accuracy (as the percent recovery) and recovery (%) are calculated (Table 3) -- what is the difference since the %recovery is the measure of accuracy? In Table 3, the plasma sample of nominal nitrite concentration of 1.3 uM was spiked with 2.1 uM nitrite, so the nominal concentration should be 3.3 uM, not 11.6 uM (the same for the 4.1 uM spike).  CV% and recovery/ accuracy in plasma and real RBC hemolysate are calculated only for 2 samples, which is insufficient for proper method validation.

4) Only LOD was calculated for method validation, while LOQ was not given. The information on the method linearity was also omitted. These parameters are essential since the linear regression equation can only be applied in a defined range. 

5) How were the confidence limits presented in Fig. 2 and 3 assessed?

6) The notation of the slope of the calibration curve in line 149 is incorrect.

7) The concentration of nitrate and nitrite was assessed using the optimized method in the real samples demonstrating method applicability. However, the way the data are presented (Fig. 6 and Fig. 7) do not provide additional information, i.e., how the extreme conditions impacted the NO2- and NO3- levels? Where they consistently lower/higher for a particular group? Etc. In fact, they are not histograms that reveal some trends in a set of data, but simple bar graphs. 

8) A brief discussion on the hemoglobin impact on the nitrate/nitrite level should be included in the RBC sample pretreatment section.

Author Response

Dear Reviewer,

thank you for the meticulous reading. Here below, please find our answer to your raised points (see attachment). We have brought several modifications to the original version to consider yours, along the other Reviewers’ comments.

Best regards

FMR, on behalf of all Authors

Round 2

Reviewer 3 Report

Thank you for the comprehensive response to my points. In general, they are satisfying, and the corrections added to the text improve the quality of the manuscript. Nonetheless, minor changes addressed below should still be considered.

The Authors have corrected the value of virtually nil intercept from (10.1±1.0*10-3) to (10.1*10-3 ± 1.0*10-3) on line 153; however, the value of the slope is still incorrect. It is stated that the slope of the calibration curve for the nitrite is 4.048 ± 0.011*10-3, while according to the presented figures and data notations in the latter text, it should be 4.048*10-3 ± 0.011*10-3.

In my opinion, the Authors should not call Figures 6 and 7 histograms since it is misleading. A histogram is a bar graph-like representation of data that groups a range of outcomes into columns along the x-axis. Here, simple bar graphs are presented that show the ordered distribution of measured values in the pilot test, not histograms. I would suggest modifying the captions, e.g., by omitting the type of the graph used, i.e., “The measured nitrite (blue bar) and nitrate (red bar) levels in 100 plasma samples…”.

There is still some inconsistency in data presentation. Table 3 reports the accuracy, precision, and recovery of added nitrite and nitrate to human plasma, and Table 5 reports the same values for human red blood cells hemolysate. In the fourth column, the accuracy is given in ± %, but different methodologies had to been used since values close to 100 % and ca. or below 1% are given. Using the same formula for accuracy calculation would be beneficial for the paper's clarity. Also, the description of the spiked samples should be reconsidered since it is ambiguous. As an example, in Table 3, samples have been spiked with either 10.4 or 20.8 uM of NO2-. This notation suggests that 10.4 or 20.8 uM of NO2- has been added to the original sample. As indicated in the spreadsheet kindly provided by the Authors, a smaller amount of nitrite has been added, i.e., 10.3 or 20.6 uM (10,4 nmol of NO2- have been added to the plasma sample= 10 uL*1,04 mM). When using the final concentration of the nitrate/ nitrite standard added in the description, the numbers would match up, and thus the confusion would be avoided.

Author Response

Dear Reviewer 3, thank you very much for this additional tier of debug. All your notes have been duly corrected. We gave the text a further run of English editing with the Grammarly software. We hope tha now everything is good.

Best regards.

FMRubino, on behalf of all Authors